# Dependence of learning outcomes in flipped and lecture classrooms on review questions: A randomized controlled trial and observational study

Jason Pitt ᴰ*, Bethany Huebner

Department of Physical Therapy, University of Evansville, Evansville, Indiana, United States of America

* jp351@evansville.edu

## Abstract

### Purpose

The effects of class structure (i.e., lecture vs. flipped) and repeated review on: 1) exam scores and 2) student perceptions of learning were compared in a prospective randomized controlled trial (Study 1) and a retrospective cohort study (Study 2).

### Methods

In Study 1, 42 second year students in a Doctor of Physical Therapy program were randomized to either a lecture or flipped section of a neurobiology class. Both sections incorporated repeated review. In Study 2, exam scores were retrospectively compared between two cohorts: a lecture cohort without repeated review (n = 42) and a flipped cohort with repeated review (n = 46). In both studies, outcomes of interest were exam scores and student surveys.

### Results

In Study 1, students in the lecture and flipped sections had similar exam averages (lecture = 76.7 ± 17%, flipped = 77.5 ± 17%, p = 0.73). Repeated review significantly improved exam scores by 12.0 percentage points (95% CI: 8.0 to 16.0 percentage points) in the lecture section and 10.8 percentage points (95% CI: 6.9 to 14.8 percentage points) in the flipped section. The flipped section reported higher levels of satisfaction and perceived learning. In Study 2, the flipped cohort had significantly higher exam scores than the lecture cohort (lecture = 70.2 ± 6.9%, flipped = 83.4 ± 7.7%, p < 0.0001). Student satisfaction and perceived learning were similar in both cohorts.

### Conclusion

Exam scores improve with review questions and quizzes provided in a class, both in a lecture or flipped classroom.

**Data Availability Statement:** The minimal anonymized data sets necessary to replicate our study findings are available at the following URL: https://osf.io/qw4x9/.

**Funding:** The authors received no specific funding for this work.

**Competing interests:** The authors have declared that no competing interests exist.

## Introduction

There is growing enthusiasm for the use of flipped classes in health professions education [1–12]. On Pubmed, over 100 articles on flipped classes were indexed per year in 2020 and 2021, compared to fewer than ten articles indexed per year before 2014. In flipped classes, lecture material is delivered before class, and class time is used for discussion of review questions, simulation, or application of skills. Consistent evidence indicates that students perceive flipped classes to be more enjoyable and effective at promoting knowledge retention [11–15].

All flipped classrooms involve some element of repeated review (e.g., graded quizzes, electronic learning modules, or class discussion). We broadly define repeated review as any activity that requires students to answer questions related to class material, as that is the same task performed on exams. Repeated review improves learning outcomes in undergraduate and graduate medical students [16–18]. Furthermore, repeated review is efficacious across age groups, from first graders to elderly patients with dementia [19, 20], and improves exam scores more than repeated study or other active learning tasks such as concept mapping [21].

Despite educator enthusiasm, flipped classrooms provide inconsistent improvements in objective measures of knowledge retention (i.e., exam scores). A meta-analysis focusing on health professions education reported significant improvements on class exams and clinical exams in flipped classrooms [22]. However, this effect was lost when only randomized studies were included in follow-up moderator analyses. Interestingly, the standardized mean difference between flipped classrooms and the control group was greater when: 1) assessments and exercises were made available to students and 2) quizzes were given at the beginning of each class, suggesting that learning outcomes depend on repeated review [22]. Consistent with this notion, only one of the four randomized studies included in the meta-analysis reported a significant difference between the flipped and lecture sections [23]. In this study, students in the flipped section completed additional exercises during class (i.e., were exposed to additional review questions). The other three studies reported no significant difference in objective learning outcomes and did not provide additional review questions to the flipped section [24–26]. From these data, we hypothesized that exam scores in flipped and lecture classes would be: 1) similar if both classes were provided the same review questions, and 2) higher in class sections that receive a greater number of review questions.

To address these hypotheses, we conducted two studies. Study 1 was a semester-long randomized controlled trial comparing multiple-choice exam scores and student perceptions of learning between students in a flipped class and students in a lecture class with an equivalent degree of review questions. We sought to estimate how exam scores were impacted by the flipped class structure and availability of review questions. Study 2 was a follow-up observational study comparing exam scores between two different student cohorts: a lecture cohort and a flipped cohort. The purpose of the second study was to determine whether providing additional review questions increases exam scores. The primary analysis compared a lecture cohort to a flipped cohort with additional review to recreate the experimental set-up of many flipped class studies. A secondary analysis was conducted to test the importance of review questions independent of class format. In the secondary analysis, we compared: 1) the flipped section from study 1 to the flipped cohort from study 2, which received additional review and 2) the lecture cohort from study 2 to the lecture section from study 1, which received additional review.

## Materials and methods

### Flipped class development

In fall 2018, the course was taught as a lecture class without repeated review. In fall 2019, the course was restructured as two separate sections: a lecture section and a flipped section. Both

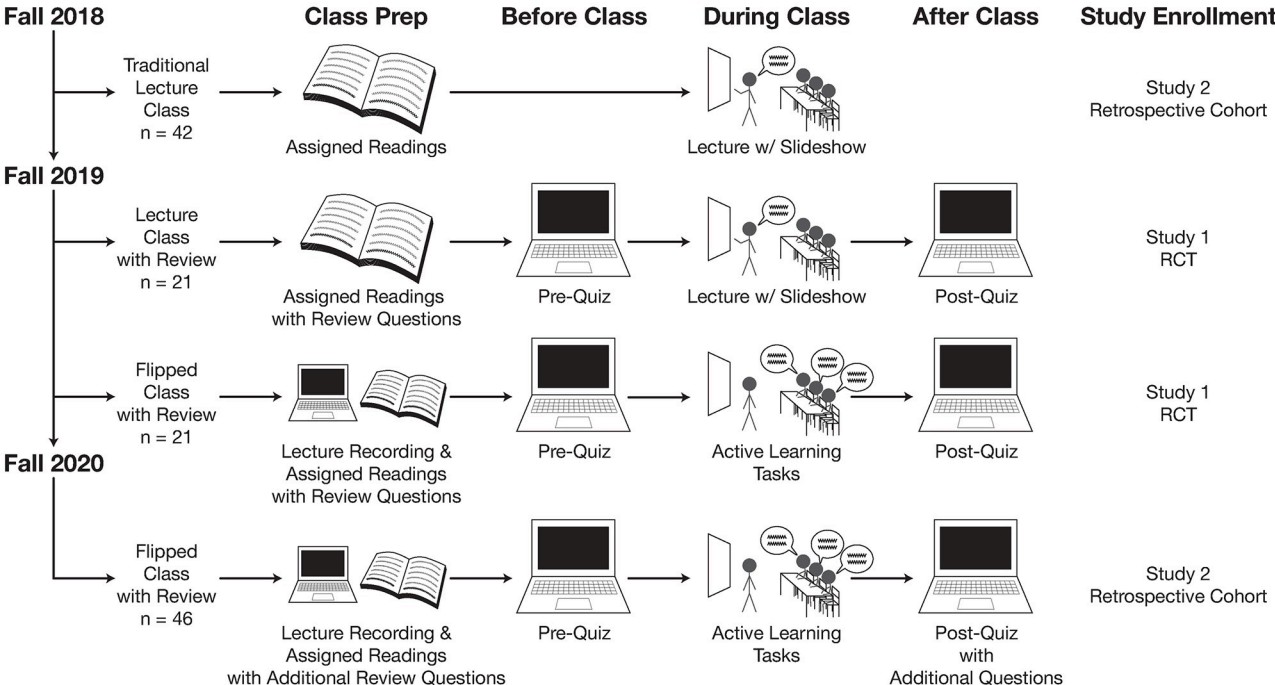

**Fig 1. Development of the flipped class.** In fall 2018, the course was structured as a lecture class. In fall 2019, the course was divided into two sections: a lecture class with review and a flipped class. Both class sections included the same review questions provided as: 1) open-ended review questions in the reading notes, 2) true-false questions in pre-quizzes, and 3) multiple-choice questions in post-quizzes. In fall 2020, the class was structured as a flipped class with additional review questions and new lecture recordings.

sections included repeated review. In fall 2020, the course was taught as a flipped class with additional review questions compared to 2019. Fig 1 summarizes the class structure in each academic year.

## Study 1

**Setting and participants.**  Study 1 was a prospective randomized controlled trial carried out in a neurobiology class during the fall semester of 2019 (Fig 2). Forty-two second year students (10 males, 32 females, age = 23.53 ± 1.26 years) in the University of Evansville's Doctor of Physical Therapy program were enrolled. A prospective power analysis indicated that our sample size of 42 could detect a difference of 4.8 percentage points with 80% power. Although blinding was not possible, all exams were multiple-choice and graded electronically to minimize bias. Student questionnaires were administered electronically and completed in the absence of either researcher.

**Ethics statement.**  The study was reviewed by the Institutional Review Board at the University of Evansville and deemed exempt [45 CFR 46.104(d)]. On the first day of class, the instructor explained the goals of the project and obtained written informed consent from all students before enrollment in the study. After each participant and a witness signed the consent form, all forms were collected and stored in the principal investigator's office.

**Randomization.**  Students were randomized 1:1 to either a standard lecture section or a flipped section using a random number generator—neither researcher played a role in subject allocation.

**Class format.**  *Lecture Section.* The lecture section met Tuesday and Thursday. Before each class, students completed: 1) assigned readings and 2) a true/false quiz. Assigned readings

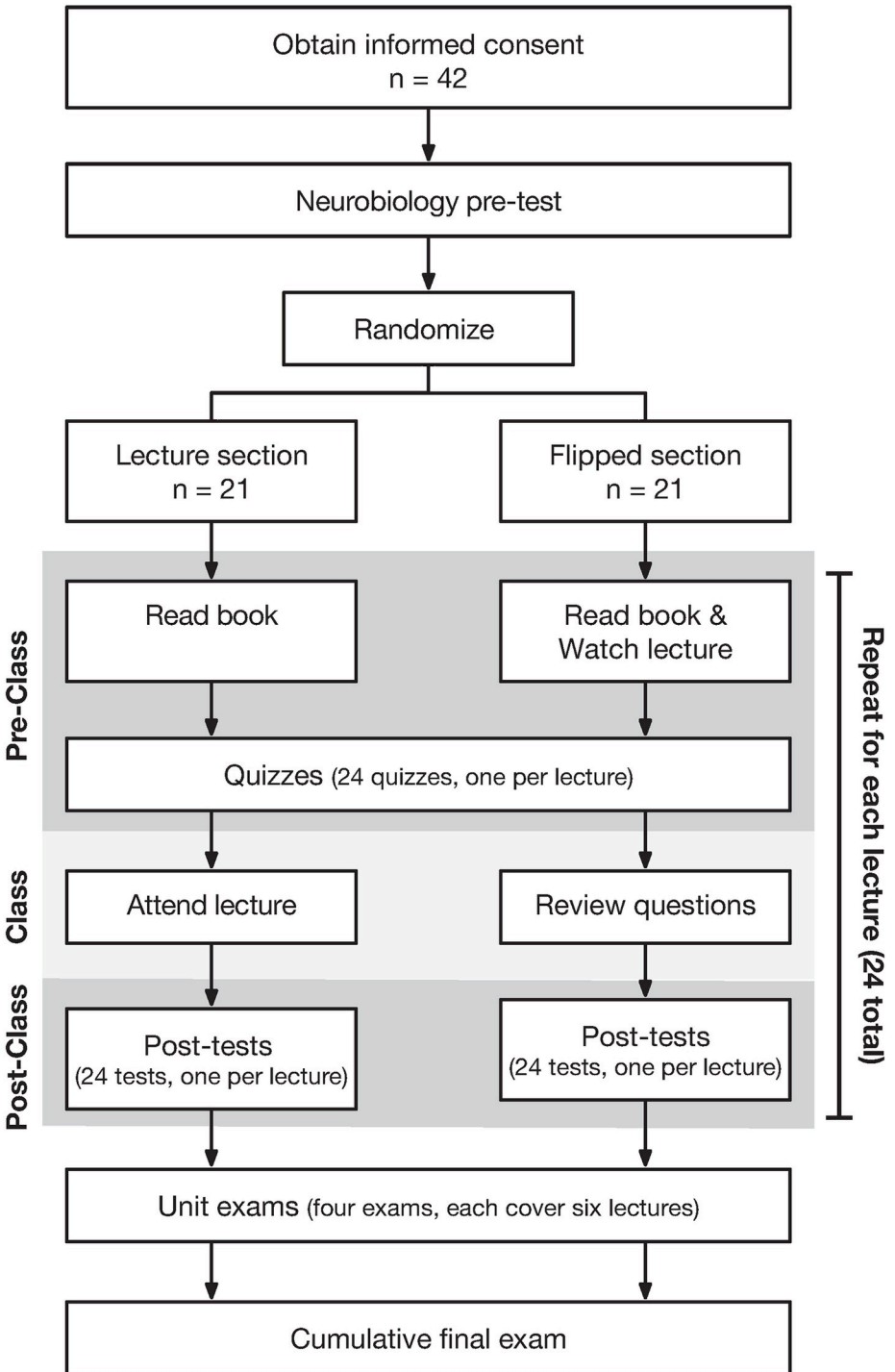

**Fig 2. RCT experimental flow chart.**

included textbook chapters and reading notes with review questions relevant to course learning objectives. Review questions spanned multiple levels of difficulty, from basic knowledge (e.g., "List the nuclei within the basal ganglia and the neurotransmitter released by their projection neurons.") to higher-level application (e.g., "Predict the motor symptoms that may

occur with damage to the different nuclei within the basal ganglia."). During class, the instructor presented a slideshow that included two breaks. During each break, the presenter: 1) answered questions from students and 2) posed a selection of review questions from the reading notes. Students were given time to discuss review questions in small groups. If time allowed, students were called on at random to answer review questions. Given the amount of information covered, it was rare to cover all review questions. After class, students completed a multiple-choice post-quiz on the material. Each lecture was recorded, with questions and small group discussion edited out, and posted to the class website, where it could be viewed by students in the lecture and flipped sections.

*Flipped Section*. The flipped section met Wednesday and Friday. Before each class, students: 1) completed reading assignments, 2) watched the lecture recording, and 3) completed the same true/false quiz as the lecture section. During class, the instructor: 1) answered questions from students and 2) posed the same selection of review questions given to the lecture section. After small group discussion, students were called on at random to answer the review questions. This form of active learning is analogous to the popular think-pair-share method. All review questions were covered in the flipped section. After class, students completed the same multiple-choice post-quiz given to the lecture section.

*Lab Section*. On Mondays, both sections met together in the same room for either: 1) lab sections or 2) an exam on lecture material. Because both sections met together, the lab section served as a negative control. Each lab section covered a specific region of the nervous system, including its: 1) location, 2) connectivity, and 3) common clinical symptoms that arise following damage. Lab sections included a slideshow and small group discussion of review questions. Material from the lab section served as the basis for the final exam.

**Outcomes measured.** A Standard Protocol Items: Recommendations for Interventional Trials (SPIRIT) checklist of assessments is shown in Table 1. Each type of assessment is described below. Our primary outcome was unit exam scores. Our secondary outcome was student surveys of perceived learning.

*Neurobiology Pre-test*. A 45-item multiple-choice exam that contained a sample of questions from the final exam was administered electronically as a secure exam in Examplify (ExamSoft).

*Pre-quizzes*. Pre-quizzes containing 6-12 true/false questions were given to encourage class preparation. Students were allowed to use the textbook and instructor notes while completing the pre-quiz. The flipped section also had access to the lecture recordings while completing the pre-quizzes.

*Post-quizzes*. Post-quizzes containing 6–12 multiple-choice questions were completed within 24 hours after each lecture to expose students to graded review questions related to lecture material. Questions were similar, but not identical, to exam questions. Students in both sections were allowed to use the textbook, instructor notes, and lecture recordings while completing the post-quizzes.

*Exams*. Four exams containing 40–45 multiple-choice questions were administered electronically as a secure exam in Examplify. Students in both sections took each exam at the same time in the same room. Students were not allowed to use any outside sources while taking exams.

*Final Exam*. A final exam containing 50 questions from the lab section was administered electronically as a secure exam in Examplify. Students were not allowed to use any outside sources while taking the final exam.

*Class Satisfaction Survey*. A nine-item Likert scale survey (1 = strongly disagree, 5 = strongly agree) that measured student motivation, perceived learning, and satisfaction with the class was given after the first exam (five weeks) and before the final exam (15 weeks). Survey items are listed in Table 4.

**Table 1. Standard Protocol Items: Recommendations for Interventional Trials (SPIRIT) checklist for Study 1.**

| | STUDY PERIOD | | | | | | | | | |
| --- | --- | --- | --- | --- | --- | --- | --- | --- | --- | --- |
| | Enrollment | Allocation | Post-allocation | | | | | | | Close-out |
| Time point | Day 0 | Day 1 | Day 5 | Day 26 | Day 35 | Day 54 | Day 75 | Day 103 | Day 105 | Day 110 |
| **Enrollment:** | | | | | | | | | | |
| Eligibility screen | X | | | | | | | | | |
| Informed consent | X | | | | | | | | | |
| Allocation | | X | | | | | | | | |
| **Interventions:** | | | | | | | | | | |
| Lecture section | | | | | | | | | | |
| Flipped section | | | | | | | | | | |
| **Assessments:** | | | | | | | | | | |
| GPA calculations | X | | | | | | | | | |
| Neurobiology pre-test | | X | | | | | | | | |
| Pre-quiz | | | | | | | | | | |
| Post-quiz | | | | | | | | | | |
| Exam | | | | X | | X | X | X | | |
| Final exam | | | | | | | | | | X |
| Satisfaction survey | | | | | X | | | | X | |

*IOTA360 Course Evaluation*. The IOTA360 course evaluation is a Likert scale survey (1 = strongly disagree, 5 = strongly agree) given in all courses offered at the university.

## Study 2

**Setting and participants.** Study 2 was a retrospective analysis of exam scores in the same neurobiology class from 2018 and 2020. In 2018, the course was structured as a lecture class and contained 42 students. In 2020, the course was structured as a flipped class and contained 46 students. Exam scores were exported from the course gradebooks for retrospective analysis.

**Class format.** *Fall 2018 Lecture Cohort*. In fall 2018, students prepared for lecture by completing assigned readings, which included textbook chapters and notes from the professor. During class, the professor delivered a slideshow presentation to cover the material. No quizzes were given after class to review the lecture material.

*Fall 2020 Flipped Cohort*. In fall 2020, the course structure was nearly identical to that of the flipped section in fall 2019. In 2020, additional review questions were used compared to previous years; however the types of questions still spanned multiple levels of difficulty. Multiple-choice post-quizzes contained a greater number of questions than in 2019.

**Outcomes measured.** *Exams.* Four exams containing 40–45 multiple-choice questions were administered electronically as a secure exam in Examplify. Students were allowed to use a single 3"x5" note card of notes on exams in 2018. No outside sources were allowed on exams in 2020.

*IOTA360 Course Evaluation.* The same IOTA360 course evaluation administered in Study 1 was used in Study 2.

## Analysis of the outcomes

Data analysis and plotting were carried out in RStudio (version 1.4.1106, R version 4.0.5). Descriptive statistics are reported as average ± standard deviation. A p-value of $< 0.05$ was considered significant for all inferential tests. Nominal data were compared using Pearson's Chi-squared tests. Continuous data were compared using Welch two sample t-tests. The effects of repeated review, class format, and their interaction on exam scores were examined using ordinary least-squares regression with the following covariates: age, undergraduate GPAs, graduate GPA, and neurobiology pre-test score. Likert scale scores from student questionnaires were compared using Wilcoxon rank sum tests with continuity correction. Boxplots represent the median (line), first and third quartiles (lower and upper hinges), and the minimum and maximum values (lower and upper whiskers, not exceeding 1.5 interquartile ranges).

## Results

### Study 1

**Academic outcomes: Effect of class format.** Students in the lecture and flipped sections were well-matched academically (Table 2). Academic performance throughout the semester was nearly identical in the lecture and flipped sections (Fig 3). As expected, pre-quiz scores were significantly higher in the flipped section (lecture = 86.8 ± 2.7%, flipped = 89.2 ± 4.6%, p = 0.04)—students in the flipped section could watch lecture recordings while taking pre-quizzes. Post-quiz scores were similar in both sections (lecture = 81.9 ± 9.2%, flipped = 78.0 ± 6.2%, p = 0.09). Completion rates were high for both pre-quizzes (lecture = 99.8%, flipped = 99.6%, p = 1) and post-quizzes (lecture = 96.8%, flipped = 98.2%, p = 0.22). Average exam scores were similar in both sections (lecture = 76.7 ± 17%, flipped 77.5 ± 17%, p = 0.64). All students met synchronously for the lab sections, allowing the final exam to act as a negative control. As expected, final exam scores were similar in the two sections (lecture = 83.0 ± 6.7%, flipped = 82.2 ± 7.0%, p = 0.73).

**Table 2. Baseline Characteristics of RCT Cohort (Fall 2019).**

| Outcome | Study 1: Randomized-Controlled Trial | | | Study 2: Retrospective Analysis | | |
| --- | --- | --- | --- | --- | --- | --- |
| | Lecture group | Flipped group | P value | Lecture cohort | Flipped cohort | P value |
| Sample size | 21 | 21 | | 42 | 46 | |
| Male:female | 5: 16 | 5: 16 | p = 1 | 10: 32 | 11: 35 | p = 1 |
| Age (years) | 23.5 (1.4) | 23.5 (1.2) | p = 0.93 | 23.5 (0.84) | 23.4 (0.84) | p = 0.66 |
| Undergraduate cumulative GPA | 3.77 (0.13) | 3.72 (0.16) | p = 0.29 | 3.71 (0.17) | 3.7 (0.16) | p = 0.92 |
| Undergraduate science GPA | 3.55 (0.32) | 3.49 (0.31) | p = 0.54 | 3.41 (0.38) | 3.47 (0.32) | p = 0.43 |
| Graduate GPA | 3.59 (0.24) | 3.58 (0.25) | p = 0.91 | 3.69 (0.18) | 3.74 (0.18) | p = 0.17 |
| Neurobiology pre-test score (%) | 45.48 (8.35) | 44.52 (8.57) | p = 0.72 | - | - | - |

Male:female ratios are reported as counts and compared between groups using Pearson's Chi-squared test with Yates' continuity correction. All other values are reported as mean (standard deviation) and compared between groups using Welch's t-tests.

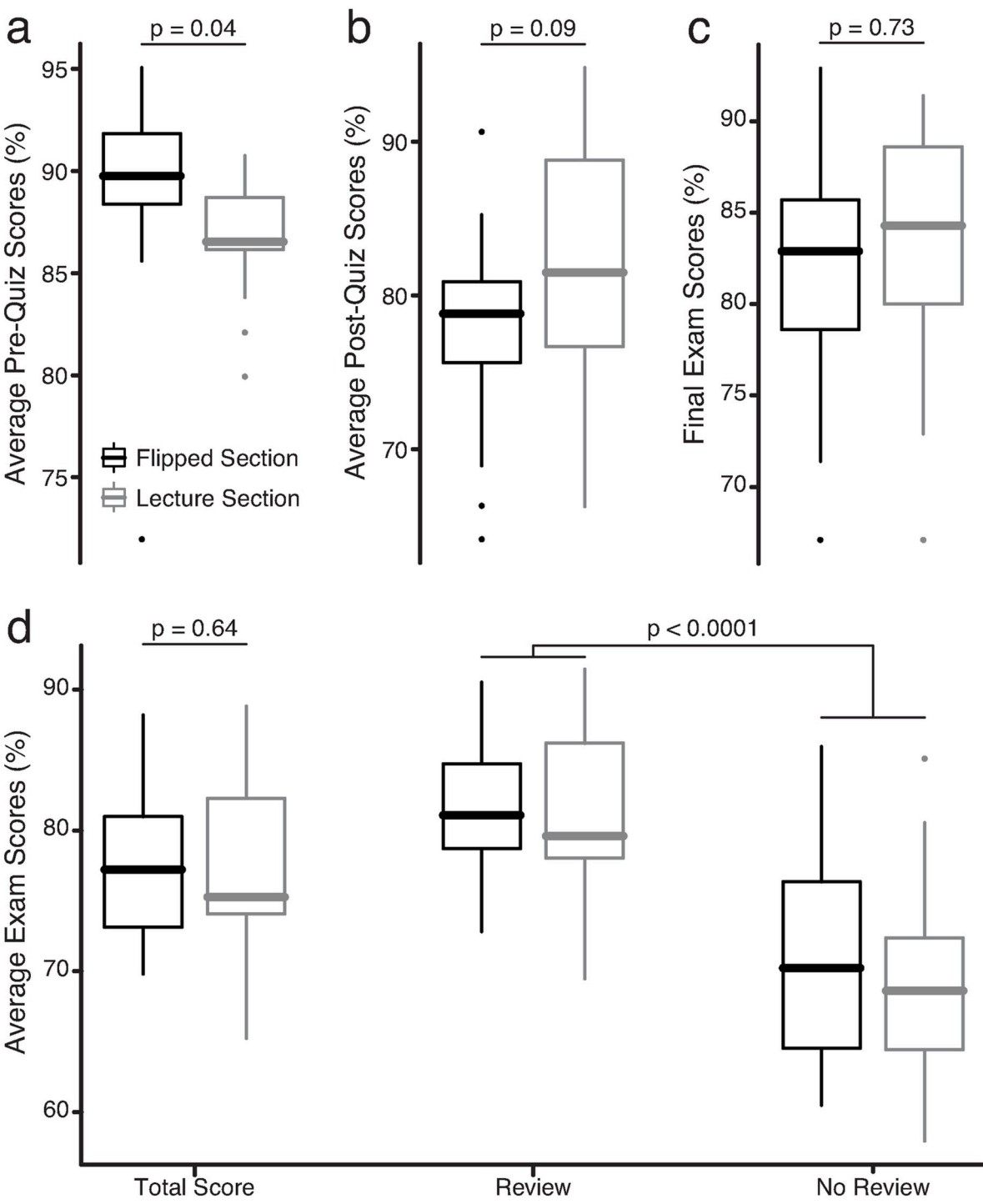

**Fig 3. Performance on quizzes and exams.** (a) Pre-quiz scores were higher in the flipped section (black) than the lecture section (gray). Pre-quizzes were prospective for the lecture section and retrospective for the flipped section. No significant differences were found between scores in the flipped and lecture sections on post-quizzes (b) or the final exam (c). (d) Total exam scores were similar in the flipped and lecture sections. Scores on exam items that were based on review or quiz questions (Review) were significantly higher than scores on novel exam questions (No Review). This effect was consistent in the flipped and lecture sections. P-values were obtained from Welch's t-tests.

**Table 3. Exam scores in lecture and flipped groups.**

| Exam | Review type | Unadjusted exam scores, mean (SD) | | Adjusted mean differences (95% CI)[a] | | | |
|---|---|---|---|---|---|---|---|
| | | Lecture group (n = 21) | Flipped group (n = 21) | Lecture vs. flipped | P value | Reviewed vs. Unreviewed | P value |
| Average[b] | Total Score | 76.7 (6) | 77.5 (5.4) | -1.5 (-4.7 to 1.7) | 0.341 | 10.8 (7.7 to 14) | <0.0001 |
| | Reviewed | 80.8 (6.1) | 81.4 (5) | | | | |
| | Unreviewed | 68.8 (6.7) | 70.6 (7.4) | | | | |
| 1 | Total Score | 80.6 (7.4) | 87.1 (5.6) | -10.5 (-15.6 to -5.3) | 0.0001 | 1.8 (-3.2 to 6.9) | 0.4705 |
| | Reviewed | 82 (8.9) | 87.6 (5.6) | | | | |
| | Unreviewed | 75.7 (9) | 85.7 (11.2) | | | | |
| 2 | Total Score | 69.7 (9.2) | 68.4 (10.9) | 2.4 (-4.4 to 9.2) | 0.4829 | 7.3 (0.6 to 14) | 0.0337 |
| | Reviewed | 72.2 (9.5) | 71.4 (12.1) | | | | |
| | Unreviewed | 66.4 (11.7) | 64.2 (13.7) | | | | |
| 3 | Total Score | 81.2 (8) | 80.3 (7.6) | 2.3 (-3.9 to 8.5) | 0.4583 | 18.3 (12.2 to 24.4) | <0.0001 |
| | Reviewed | 86.4 (7.7) | 86.1 (8.8) | | | | |
| | Unreviewed | 69.6 (13) | 67.8 (12.1) | | | | |
| 4 | Total Score | 75.1 (6.2) | 74.2 (7.7) | -0.4 (-5.5 to 4.7) | 0.8721 | 16 (10.9 to 21) | <0.0001 |
| | Reviewed | 82.7 (7.7) | 80.6 (8.2) | | | | |
| | Unreviewed | 63.7 (9) | 64.6 (10.5) | | | | |

[a] Mean differences in exam scores were adjusted for age, sex, neurobiology pre-test scores, undergraduate GPAs, and graduate GPAs using the following formula: exam score ~ age + sex + neurobiology pre-test + sGPA + cGPA + gGPA + review type * class section.

[b] Average exam scores were calculated for each student as the mean of exams 1–4.

**Academic outcomes: Effect of repeat review.** To measure the effects of repeated review on exam scores, exams were written to include a mix of questions that were either: 1) based on review or quiz questions (called "reviewed") or 2) based on lecture material, but with no review questions provided (called "unreviewed"). Scores were significantly higher for reviewed questions (Fig 3d). On average, students scored about a letter grade higher on reviewed questions (reviewed = 81.1 ± 13%, unreviewed = 69.7 ± 13%, p < 0.0001). Repeated review increased exam scores similarly in both class sections (lecture = 12.0 ± 5.0%, flipped = 10.9 ± 4.7%, p = 0.45).

Multiple linear regression was used to control for differences in age, sex, graduate GPA, undergraduate GPAs, and neurobiology pre-test scores (see Table 3). Repeated review improved exam scores by 10.8% (95% CI = 7.7 to 14.0%, p < 0.0001), while enrollment in the flipped section had no significant effect on exam scores (95% CI = -1.7 to 4.7%, p = 0.34). These findings were consistent across each unit exam, except for exam 1. On exam 1, students in the flipped section had higher scores than students in the lecture section (95% CI = 5.3 to 15.6%, p < 0.0001), but repeated review did not significantly improve scores (95% CI = -3.2 to 6.9%, p = 0.47). We observed similar outcomes when we compared the unadjusted exam scores based on lecture section (lecture = 80.6 ± 7.4%, flipped = 87.1 ± 5.6%, p = 0.0025) and repeated review (reviewed = 84.8 ± 7.8%, unreviewed = 80.7 ± 11.3%, p = 0.06). On exams 2–4, exam scores were similar in the flipped and lecture sections, and repeated review significantly increased exam scores both before and after adjustment (see Table 3). Repeated review appeared to be equally effective in both class sections, as we found no significant interaction effects between class section and repeated review (exam average: p = 0.61; exam 1: p = 0.21; exam 2: p = 0.75; exam 3: p = 0.73; exam 4: p = 0.39).

**Student perceptions.** To measure class satisfaction and perceived learning, a nine-item Likert scale survey was administered at five weeks and 15 weeks (see Table 4). At both time

**Table 4. Student satisfaction survey results.**

| Item | Time point | Lecture section (n = 21) | Flipped section (n = 21) | Lecture vs. flipped |
|---|---|---|---|---|
| Total (all items) | 5 weeks | 32.65 (6.15) | 36.05 (3.63) | p = 0.04 |
| | 15 weeks | 31.76 (6.93) | 38.95 (2.78) | p < 0.0001 |
| | 5 vs 15 weeks | p = 0.83 | p = 0.005 | |
| The teaching methods used in class are helpful and effective. | 5 weeks | 3.9 (0.79) | 4.35 (0.59) | p = 0.06 |
| | 15 weeks | 3.67 (0.91) | 4.57 (0.93) | p = 0.0005 |
| | 5 vs 15 weeks | p = 0.38 | p = 0.06 | |
| The teaching methods used in this class motivate me to learn. | 5 weeks | 3.75 (0.91) | 4.1 (0.45) | p = 0.17 |
| | 15 weeks | 3.48 (1.21) | 4.29 (0.56) | p = 0.012 |
| | 5 vs 15 weeks | p = 0.69 | p = 0.23 | |
| I am provided with a variety of learning materials and activities in this class. | 5 weeks | 3.7 (1.13) | 4.55 (0.6) | p = 0.008 |
| | 15 weeks | 3.95 (0.8) | 4.76 (0.44) | p = 0.0004 |
| | 5 vs 15 weeks | p = 0.58 | p = 0.25 | |
| The provided class materials are helpful and facilitate my learning. | 5 weeks | 3.75 (0.91) | 4.4 (0.6) | p = 0.01 |
| | 15 weeks | 3.9 (0.94) | 4.86 (0.36) | p = 0.0003 |
| | 5 vs 15 weeks | p = 0.71 | p = 0.006 | |
| I am confident that I am learning class content. | 5 weeks | 3.55 (0.76) | 4.15 (0.81) | p = 0.02 |
| | 15 weeks | 3.1 (0.94) | 4.24 (0.62) | p = 0.0001 |
| | 5 vs 15 weeks | p = 0.13 | p = 0.88 | |
| The way that the class is taught is enjoyable. | 5 weeks | 3.4 (0.88) | 3.8 (0.83) | p = 0.09 |
| | 15 weeks | 3.24 (1.09) | 4.14 (0.65) | p = 0.004 |
| | 5 vs 15 weeks | p = 0.68 | p = 0.21 | |
| The way that the class is taught facilitates my learning. | 5 weeks | 3.7 (0.92) | 4.2 (0.83) | p = 0.06 |
| | 15 weeks | 3.52 (0.98) | 4.67 (0.48) | p < 0.0001 |
| | 5 vs 15 weeks | p = 0.61 | p = 0.05 | |
| I learn more in class than outside of class. | 5 weeks | 3.5 (1.4) | 3.15 (1.14) | p = 0.36 |
| | 15 weeks | 3.57 (1.5) | 3.71 (1.01) | p = 1 |
| | 5 vs 15 weeks | p = 0.85 | p = 0.09 | |
| I am interested in the topics we cover in class. | 5 weeks | 3.4 (0.75) | 3.35 (0.99) | p = 0.77 |
| | 15 weeks | 3.33 (1.11) | 3.71 (0.96) | p = 0.25 |
| | 5 vs 15 weeks | p = 0.75 | p = 0.24 | |

**Notes.** All items were ranked from 1 (Strongly Disagree) to 5 (Strongly Agree). Values represent mean (standard deviation). P-values from pairwise Wilcoxon rank sum tests with continuity correction are reported.

points, survey responses had good internal consistency (Cronbach's α: 5 weeks = 0.83, 15 weeks = 0.88). Student ratings were higher in the flipped section at both five and 15 weeks for most questions on the survey. Despite nearly equivalent course grades at 15 weeks, students in the flipped class were more confident in their learning (lecture = $3.1 \pm 0.94$, flipped = $4.24 \pm 0.62$, $p < 0.0005$) and believed the teaching methods to be more effective (lecture = $3.67 \pm 0.91$, flipped = $4.57 \pm 0.93$, $p = 0.0005$). On the IOTA360, a separate institutional survey, students in the flipped section reported significantly higher ratings for course excellence (lecture = $3.75 \pm 1.02$, flipped = $4.56 \pm 0.51$, $p = 0.005$), instructor excellence (lecture = $4.1 \pm 0.85$, flipped = $4.72 \pm 0.46$, $p = 0.01$), and perceived learning (lecture = $4.05 \pm 1$, flipped = $4.67 \pm 0.49$, $p = 0.04$).

## Study 2

**Academic outcomes: Effect of class format.** To match the experimental approach used in many flipped class studies, we retrospectively compared exam scores between two cohorts:

1) a lecture class with no review questions or quizzes and 2) a flipped class with review questions and quizzes. Both cohorts had similar undergraduate and graduate GPAs (see Table 1). The flipped cohort had significantly higher exam averages than the lecture cohort (lecture = 70.2 ± 6.9%, flipped = 83.4 ± 7.7%, p < 0.0001).

**Academic outcomes: Effect of repeated review.** Between 2019 and 2020, additional review questions were created and provided to students. Interestingly, exam scores were significantly higher in 2020 than the flipped section of 2019 (2019 = 77.5 ± 5.4%, 2020 = 83.4 ± 7.7%, p = 0.002). Both sections were flipped classes that viewed the same lecture videos before class and completed the same exams. The difference in exam performance may be explained by the additional review and quiz questions provided in 2020. In 2019, only 65% of exam questions had a related review or quiz question compared to 80% in 2020 ($\chi^2(1)$ = 9.03, p = 0.003). Similarly, exam scores were also significantly higher in the 2019 lecture section than the 2018 lecture cohort (2019 = 76.7 ± 6.0%, 2020 = 70.2 ±6.9%, p = 0.0004).

**Student perceptions.** Despite differences in learning outcomes between the 2018 and 2020 cohorts, IOTA360 student evaluations of the course were similar in both cohorts. For example, students provided similar ratings of course excellence (lecture = 4.28 ± 0.99, flipped = 4.12 ± 1.03, p = 0.80), instructor excellence (lecture = 4.45 ± 0.95, flipped = 4.56 ± 0.67, p = 0.28), and perceived learning (lecture = 4.38 ± 0.78, flipped = 4.49 ± 0.8, p = 0.52).

## Discussion

The purpose of this study was to measure and compare the effects of repeated review on exam performance in traditional lecture and flipped class sections. In a randomized trial, exam performance improved as a function of repeated review and was not related to class format. The flipped class was not more effective than a lecture class with equivalent repeated review. Although no academic differences were observed between the two sections, students in the flipped section reported a higher degree of perceived learning and course satisfaction. In a retrospective observational study, exam performance also improved as a function of repeated review. Exam scores were significantly higher in a flipped cohort that received repeated review compared to a lecture cohort that did not receive repeated review. Exam scores were also significantly higher in the flipped cohort compared to the flipped section of the randomized trial, which received fewer review questions. Despite significant differences between exam scores in the lecture and flipped cohorts, students reported comparable levels of perceived learning and class satisfaction. From this, it appears that exam performance improves as a function of repeated review but not class format.

### Effect of class format

Compared to lecture classes, flipped classes may improve average exam scores when they provide additional review questions to students. We observed this in our second study, showing that the flipped cohort had higher exam averages than the lecture cohort, which received no repeated review. In the literature, flipped classes provide significant improvements in knowledge retention when compared to traditional lecture classes that do not include repeated review [10, 15, 23, 27–30]. In these studies, both class format and the amount of repeated review differ between the groups. Although the academic improvements were attributed to differences in class format, we believe that differences in repeated review better explain the findings. For example, academic differences also occur when two flipped classes are compared—so long as one class receives additional review. In Study 2, we found significantly higher exam scores in the flipped cohort of 2020 compared to the flipped section of 2019. Both groups watched the same lecture recordings and read the same reading assignments, but the 2020

cohort received additional review questions. Similar findings were reported in a biochemistry course for undergraduate medical students [31]. The course was provided as a flipped course in two sections: one section received additional reading, and the other section received multiple-choice review questions. As in our study, both flipped sections watched the same recordings, but the section that received additional review questions had significantly higher exam scores. Thus when class format is held constant and repeated review differs between groups, academic differences are noted.

On the other hand, when class format differs between groups and repeated review is held constant, students in flipped and lecture sections perform equally on exams. In Study 1, the flipped and lecture sections received the same amount of review questions, and had similar exam averages. This finding is consistent with randomized flipped class studies that control for differences in repeated review [24–26, 32–34]. Specifically, in two courses within an ophthalmology clerkship training program, no statistically-significant differences in exam scores were found between students randomized into: 1) a lecture section that provided review questions as homework or 2) a flipped section that used the review questions for in class discussion and presentations [24]. Additionally, in a first-level medical-surgical nursing theory course, students were randomized into flipped or lecture sections that completed the same quizzes, exams, and written paper [25]. No differences were found on any outcome measure, including total exam scores, knowledge-level exam questions, or application-level exam questions. Finally, in a multi-campus randomized trial, medical students in an evidence-based medicine (EBM) class were randomized into: 1) a didactic learning section that included a presentation and small group activities or 2) a blended learning section that included recorded lectures and in-class presentations; both sections completed critical appraisals to apply the concepts of EBM [26]. EBM competency was measured using the Berlin Questionnaire and ACE tool, and no significant differences were found between the two sections. Our work extends these findings by estimating the impact of review questions on exam scores and demonstrating that the effect of repeated review was similar in lecture and flipped sections. From this, class format appears less important to learning outcomes than including review questions.

## Effect of repeated review

Repeated review was equally effective in the flipped and lecture sections, indicating that review questions are equally beneficial when given inside or outside of class. This is supported by findings from studies carried out in classroom settings. Including daily, in-class quizzes into two large Introductory Psychology courses (n = 982) raised average exam scores by 5.9% points compared to performance on identical questions in previous semesters [18]. In a General Biology course intended for undergraduate students planning careers in the health sciences, a large sample of >1800 students were given unlimited access to an online database of 1020 multiple-choice review questions outside of class [17]. Average exam scores were 6.6% points higher in the sections with review questions compared to historic performance. The improvements reported in both of the previous studies are consistent with our findings. In Study 1, we found that the repeated review was equally effective in the flipped section (i.e., in class) and lecture section (i.e., outside of class). In Study 2, introducing review questions to the lecture section in 2019 improved average exam scores by 6.5% points, which is comparable to the 5.9% and 6.6% point improvements reported in other studies [17, 18].

The argument that the effect was due to memorization of specific questions can be ruled out, as no review or quiz questions were repeated on the exams. For example, the following review question was given during class: "Define driving force, and compare the driving force for $K^+$, $Na^+$, and $Cl^-$ under resting conditions." The related exam question was: "When is the

driving force for an ion 0 mV?" Because we did not include any of the review questions on exams, repeated review likely improved students' understanding of class concepts, rather than specific questions. If repeated review enhances understanding of concepts, then educators may consider tailoring their formative assessments to align with difficult class concepts. For example, if students consistently struggle with a particular class concept, then additional review questions on the concept could be provided in subsequent offerings of a course.

## Student perceptions

Students believe that flipped classes are more effective than lecture classes. While exam scores were equivalent in both sections of fall 2019, students in the flipped section gave consistently higher course ratings than students in the lecture section. This difference was reliably observed on two separate class surveys and matches the greater degree of class satisfaction reported in several other flipped class studies [11–15]. Ratings of perceived learning were higher in the flipped section, and some students in the lecture section raised concerns about differences in perceived learning during office hours and after class. This inconsistency between perceived learning and learning measured objectively by exam scores has been reported by other groups [21, 35].

Interestingly, the perceived learning discrepancy only appeared in Study 1 when students were randomly assigned to one of two groups. Importantly, students were able to view a de-identified gradebook at any time during the semester. Thus students in the flipped and lecture sections could compare their scores to the class average. Although both class sections had nearly identical grade distributions, students in the lecture section believed that they were receiving an inferior educational experience and had lower scores than the flipped section. In Study 2, all students in each cohort received the same course delivery, and there were no differences in perceived learning between the lecture and flipped cohorts. The difference in perceived learning and course quality reported in Study 1 are consistent with the broader literature on the effects of blinding on subjective outcome measures. For example, multiple meta-analyses report that subjective outcomes are exaggerated in studies with unclear or inadequate blinding [36–38]. This highlights the importance of providing objective measures of learning when comparing different teaching approaches. Studies that rely on subjective outcomes are likely to provide biased effect estimates, especially given the inability to blind subjects in educational studies.

## Limitations

There are three key limitations to our trial. First, this study was carried out in a single class taught by a single professor, which limits the generalizability of our results. Other professors may adopt different flipped class models that vary in teaching effectiveness. Furthermore, flipped classroom effectiveness may vary based on a spectrum of student cognitive and non-cognitive traits, such as: educational background, motivation, study skills, and learning styles. Second, students in Study 1 were not blinded. Knowledge of their allocation appears to have influenced their survey ratings. Although course ratings were similar in 2018, 2019, or 2020, ratings in 2019 were significantly lower in the lecture section, and many students in the lecture section believed that their section had lower grades than the flipped section. From this, student ratings appear to have a high risk of bias and should be interpreted with caution. Third, the course was purely lecture-based. It is of considerable interest to determine whether flipped classes improve learning outcomes in kinesthetic or laboratory courses by affording more time for hands-on training.

### Future research

Our findings and the literature indicate that differences in learning outcomes in lecture and flipped class sections are amplified when one class section receives additional review questions (e.g., in-class discussion or graded quizzes). Based on this, we hypothesize that differences in repeated review may explain the inconsistent academic improvements reported in flipped classroom studies. The degree to which differences in repeated review explains the heterogeneity within the flipped classroom literature should be explored further in a systematic review and meta-analysis.

Given the established effectiveness of repeated review, it should be of considerable interest to optimize its use to improve retention of class concepts. Several factors likely contribute, including the number, format, and difficulty of review questions. High-quality, randomized studies should identify optimal parameters for incorporating repeated review in classroom settings. Additionally, follow-up studies should be carried out to determine whether increasing the degree of repeated testing in an introductory course improves exam performance or critical thinking in an advanced course.

## Conclusion

Based on our findings we recommend utilizing repeated review in whatever teaching environment works best for the instructor and the course content. However, educators teaching in graduate health professions education may consider using flipped classrooms since students at this level prefer this type of learning environment.

## Author Contributions

**Conceptualization:** Jason Pitt.

**Formal analysis:** Jason Pitt.

**Methodology:** Jason Pitt.

**Supervision:** Bethany Huebner.

**Visualization:** Jason Pitt.

**Writing – original draft:** Jason Pitt, Bethany Huebner.

**Writing – review & editing:** Jason Pitt, Bethany Huebner.

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
