## [Decision Letter · Decision Letter 0]

17 Oct 2022

PONE-D-22-20922Dependence of learning outcomes in flipped and lecture classrooms on review questions: a randomized controlled trial and observational studyPLOS ONE

Dear Dr. Jason Pitt,

Thank you for submitting your manuscript to PLOS ONE. After careful consideration, we feel that it has merit but does not fully meet PLOS ONE’s publication criteria as it currently stands. Therefore, we invite you to submit a revised version of the manuscript that addresses the points raised during the review process.

ACADEMIC EDITOR: 1. As the study is randomized controlled trial, authors are requested to comment on Trial Registry &    Institutional Ethical Committee approval.2. As per the policy of journal, Authors are required to make all data underlying the findings described fully available, without restriction.3. Authors are requested to reply to queries raised by Reviewer #1 & # 2.

We look forward to receiving your revised manuscript.

Kind regards,

Priti Chaudhary, M.S.

Academic Editor

PLOS ONE

Journal Requirements:

Reviewers' comments:

Reviewer's Responses to Questions

**Comments to the Author**

1. Is the manuscript technically sound, and do the data support the conclusions?

Reviewer #1: Yes

Reviewer #2: Yes

2. Has the statistical analysis been performed appropriately and rigorously? 

Reviewer #1: I Don't Know

Reviewer #2: Yes

3. Have the authors made all data underlying the findings in their manuscript fully available?

Reviewer #1: No

Reviewer #2: Yes

4. Is the manuscript presented in an intelligible fashion and written in standard English?

Reviewer #1: Yes

Reviewer #2: No

5. Review Comments to the Author

Reviewer #1: The study entitled "Dependence of learning outcomes in flipped and lecture classrooms on review

questionsis" is well conducted and written. However, Discussion needs to be more elaborate as its the heart of the study. Add some relevant references and discuss your findings more extensively.

Reviewer #2: To authors,

I am very happy to review your manuscript.

You showed the efficacy on the flipped cohort learning outcomes. It may seemed to be informative and helpful for the education. However, there are fatal problems.

One of the biggest problem is that your results have had no power to make feel something new, "NOIES".

6. PLOS authors have the option to publish the peer review history of their article (what does this mean?). If published, this will include your full peer review and any attached files.

Reviewer #1: No

Reviewer #2: No

---

## [Author Response · Author response to Decision Letter 0]

28 Nov 2022

We thank you for your constructive feedback on our manuscript. We have addressed all comments regarding: 1) formatting, 2) data availability, 3) IRB exemption, and 4) description of informed consent. The Discussion section was rewritten to address feedback from Reviewer 1. Thank you for considering our manuscript for publication. We appreciate your time and look forward to your response.

---

## [Editor Report · Decision Letter 1]

5 Dec 2022

Dependence of learning outcomes in flipped and lecture classrooms on review questions: A randomized controlled trial and observational study

PONE-D-22-20922R1

Dear Dr. Jason Pitt

We’re pleased to inform you that your manuscript has been judged scientifically suitable for publication and will be formally accepted for publication once it meets all outstanding technical requirements.

Kind regards,

Priti Chaudhary, M.S.

Academic Editor

PLOS ONE
---

## [Editor Report · Acceptance letter]

9 Dec 2022

PONE-D-22-20922R1 

Dependence of learning outcomes in flipped and lecture classrooms on review questions: A randomized controlled trial and observational study 

Dear Dr. Pitt:

I'm pleased to inform you that your manuscript has been deemed suitable for publication in PLOS ONE. Congratulations! Your manuscript is now with our production department. 

Kind regards, 

on behalf of

Dr. Priti Chaudhary 

Academic Editor

PLOS ONE